# OpenReview forum: "Beyond Next-Token Alignment: Distilling Multimodal Large Language Models via Token Interactions"
_ICML.cc/2026/Conference — ICML 2026 regular_

### Official Review · Reviewer_44W7 · 2026-03-08

**Soundness:** 3
**Presentation:** 4
**Significance:** 3
**Originality:** 3
**Overall Recommendation:** 4
**Confidence:** 4

**Summary:**

This paper proposes a well-motivated knowledge distillation method for multimodal large language models (MLLMs). Going beyond vanilla next-token alignment, the work explores token interactions from a novel perspective. Specifically, it focuses on Vision-Instruction Token Interactions and Intra-Response Token Interactions, introducing Instruction-aware Vision Alignment and Transition Probability Alignment, respectively. Each component is supported by sufficient analysis, and extensive comparative experiments along with a series of ablation studies demonstrate the effectiveness of the proposed method.

**Compliance With Llm Reviewing Policy:**

Affirmed.

**Final Justification:**

The authors' response has addressed most of my concerns. Therefore, I maintain my positive score.

**Key Questions For Authors:**

Overall, the reviewer holds a positive view of this work. The questions raised are centered on exploring the upper bounds of the method's extensibility, particularly its applicability to heterogeneous visual architectures and Mixture-of-Experts (MoE) based models.

**Limitations:**

Please see the Weaknesses section for more details on the major concerns regarding architectural extensibility.

**Strengths And Weaknesses:**

Strength:
1. This paper is well-written and easy to follow. The proposed method is well-motivated and technically sound. Exploring knowledge distillation through the lens of token interactions is an interesting and promising direction.
2. The authors conduct comprehensive comparative experiments and extensive ablation studies, providing empirical validation for each proposed component and design choice. This thoroughness is highly commendable.
3. The scaling analysis further demonstrates the method's generalization capability across heterogeneous model architectures, highlighting its potential for cross-architecture distillation.


Weakness:
1. While the method is well-motivated, its application appears to require certain pre-computations and analyses, such as identifying optimal layers via IRS for Vision-Instruction Token Interactions. It remains unclear whether these findings generalize across different model architectures, and whether a more adaptive selection mechanism could be developed to enhance the method's applicability and scalability.
2. The scaling analysis demonstrates knowledge distillation across different LLM architectures, validating the method's extensibility. However, the reviewer is curious about whether the approach remains effective for MLLMs with varying visual encoders, particularly when the number of generated visual tokens differs across models.
3. Whether the proposed distillation method can be extended to more advanced Mixture-of-Experts (MoE) architectures warrants further analysis and discussion.

---

> ### Author Rebuttal · Authors · 2026-03-30
>
> We sincerely appreciate your constructive comments and valuable suggestions. Thank you for the time and effort you have devoted to assessing our manuscript and helping us improve its quality. Our detailed responses to your concerns are provided below.
>
> ---
>
> > **Q1: Whether the optimal layer selection generalizes across architectures and if adaptive mechanisms could be developed.**
>
> We thank the reviewer for this insightful question. As shown in Fig. 3, the IRS trend is consistent across Qwen2-7B and Vicuna-7B, both exhibiting the same pattern of low IRS in shallow layers, peak at ~3/4 depth, and decline in final layers. This suggests that the IRS-based layer selection generalizes across different model architectures. IRS is already a data-driven adaptive mechanism that automatically identifies the most instruction-relevant layer for any given architecture, without manually specifying a fixed layer.
>
> Regarding a more dynamic selection (e.g., adapting layer selection during training): the IRS metric is defined as an expectation over data samples (Definition 1), which requires aggregating statistics across a sufficient number of samples for a reliable estimate. Integrating this computation into the training loop would introduce additional overhead and yield noisy estimates from mini-batches. In practice, IRS is computed as a lightweight one-time pre-computation (a single forward pass on a small data subset), and the selected layer remains stable across architectures, so we believe the current design strikes a good balance between adaptiveness and efficiency. We appreciate this insightful suggestion from the reviewer.
>
> ---
>
> > **Q2: whether the approach remains effective for MLLMs with varying visual encoders, particularly when the number of generated visual tokens differs across models.**
>
> We thank the reviewer for this question. **Align-TI is compatible with different visual encoders, as neither IVA nor TPA depends on the specific encoder architecture.** IVA operates on the instruction-to-vision attention map within the LLM decoder, which exists regardless of how visual tokens are produced. TPA operates solely on the output vocabulary distribution during the decoding stage and has no interaction with the visual encoder at all.
>
> Regarding the number of visual tokens, since IVA computes per-token KL divergence between teacher and student visual token distributions, the teacher and student need to produce the same number of visual tokens. In MLLM distillation, it is common to share the same visual processing pipeline between teacher and student, with the distillation focused on compressing the LLM backbone (e.g., LLaVA-KD[1], LLaVA-MoD[2], Align-KD[3]). Under this setting, the visual token count between teacher and student is inherently consistent.
>
> ---
>
> > **Q3: Whether the proposed distillation method can be extended to more advanced Mixture-of-Experts (MoE) architectures warrants further analysis and discussion.**
>
> We thank the reviewer for this question. **Since Align-TI performs distillation at output-level, it is inherently architecture-agnostic.** This is a key advantage of output-level distillation over intermediate feature-level approaches: it does not depend on the internal architecture of the model. TPA operates on output probability distributions and is independent of how the feed-forward layers are structured (dense or MoE). IVA requires only attention map access, which remains available in MoE-based models since MoE modifies the FFN layers while the attention mechanism is unchanged. Therefore, Align-TI can be directly applied to MoE-based models without any modification.
>
> ---
>
> [1] Cai, Yuxuan, et al. "LLaVA-KD: A Framework of Distilling Multimodal Large Language Models." ICCV, 2025.
>
> [2] Shu, Fangxun, et al. "LLaVA-MoD: Making LLaVA Tiny via MoE Knowledge Distillation." ICLR, 2025
>
> [3] Feng, Qianhan, et al. "Align-KD: Distilling Cross-Modal Alignment Knowledge for Mobile Vision-Language Large Model Enhancement." CVPR, 2025.

---

> > ### Author Rebuttal · Reviewer_44W7 · 2026-04-03
> >
> > Thank you for the authors' response, which addresses most of my concerns. However, I still encourage the authors to explore distillation between heterogeneous models with varying numbers of visual tokens, as this would further broaden the method's applicability and impact. I will maintain my original score.

---

> > > ### Author Response · Authors · 2026-04-03
> > >
> > > We are grateful to the reviewer for the positive feedback and for confirming that our rebuttal has addressed the concerns.
> > >
> > > Regarding the suggestion on distillation between heterogeneous models with varying visual token counts, we agree this is a valuable direction. The main technical difficulty is that IVA computes KL divergence between per-token distributions, which requires the teacher and student to share the same number of visual tokens. When token counts differ, direct alignment becomes infeasible, and designing a proper mapping strategy remains an open problem. We appreciate the reviewer for bringing this to our attention.

---

### Official Review · Reviewer_SvA6 · 2026-03-11

**Soundness:** 3
**Presentation:** 2
**Significance:** 3
**Originality:** 2
**Overall Recommendation:** 4
**Confidence:** 3

**Summary:**

This paper identifies the limitations of static next-token alignment in knowledge distillation (KD) for Multimodal Large Language Models (MLLMs) and proposes Align-TI, a novel KD framework designed from the perspective of token interactions. Align-TI consists of two core modules. First, Instruction-aware Vision Alignment (IVA) assigns importance weights to instruction-relevant visual tokens based on instruction-to-vision attention maps, thereby transferring the teacher's visual information extraction capability to the student. To select the optimal layer for this process, the authors propose the Instruction-Relevance Score (IRS). Second, Transition Probability Alignment (TPA) aligns the token-to-token transition probability matrix rather than a single next-token distribution, enabling the student to learn the teacher's dynamic autoregressive generation patterns and mitigating exposure bias. In experiments based on Qwen2/Qwen3, Align-TI-2B surpasses LLaVA-1.5-7B by a relative 7.0% and achieves state-of-the-art results compared to existing KD methods.

**Compliance With Llm Reviewing Policy:**

Affirmed.

**Final Justification:**

Based on the rebuttal, I have revised my score. In particular, the additional ablation study for W1 was helpful, and the explanation for W2 clarified my concerns regarding one-step transition alignment.

**Key Questions For Authors:**

[W1] The performance contribution of IVA is relatively small compared to TPA, and the analysis of its underlying causes is insufficient.
--> [Q1] Is it possible to provide a task-level analysis to examine whether IVA yields larger gains on specific task types where visual information dependency is high, such as TextVQA?

[W2]  TPA's empirical success under a theoretically limited one-step transition alignment lacks sufficient analysis.
--> [Q2] Can you discuss the conditions under which the method is expected to generalize beyond the benchmarks?

**Limitations:**

yes

**Strengths And Weaknesses:**

Strong points

[S1] The systematic analysis of two types of token interactions and the resulting module designs are well-motivated.

 The authors clearly distinguish between the prefilling stage (vision-instruction interaction) and the decoding stage (intra-response interaction) of MLLMs, and concretely analyze the information that existing static KD methods fail to capture at each stage. The attention weight visualizations and accumulated error analysis in Figure 2 persuasively support the necessity of IVA and TPA, respectively, and the logical flow from problem definition to solution is smooth.


[S2] The design of TPA is technically interesting, and an efficient parallel computation method is presented alongside it.

 The theoretical justification for expanding the alignment scope from O(|V|) to O(|V|²) through transition probability matrix alignment is clear, and the parallelization strategy using the ribbon attention mask drastically reduces the number of forward passes from dLN to 2N compared to a naive implementation. Furthermore, the %ExAccErr analysis in Figure 6 empirically demonstrates that TPA effectively mitigates exposure bias, deepening our understanding of the proposed method's working mechanism.


[S3] The experimental design is comprehensive, and diverse ablations clearly isolate the contribution of each component.
The paper includes SOTA comparisons across six benchmarks, comparisons with LLM-specific KD strategies (Table 3), individual and combined ablations of IVA/TPA (Table 5), IRS-based layer selection validation (Table 7), teacher/student weight comparisons (Table 8), sampling strategy comparisons (Figure 5), data/teacher/architecture scaling analyses (Section 4.4), and a fair comparison under an equivalent computational budget (Table 18)—all of which are highly thorough. In particular, the result that Align-TI-2B outperforms LLaVA-1.5-7B, a model 3.5× larger, effectively demonstrates its practical value.

---
Weak points

[W1] The performance contribution of IVA is relatively small compared to TPA, and the analysis of its underlying causes is insufficient.

 In Table 5, IVA alone achieves only a 0.8 average improvement, and its additional contribution when combined with TPA is a mere 0.3. The authors explain that IVA operates "indirectly," but there is no in-depth analysis of why visual token alignment has such a limited impact on response quality. Moreover, the 0.6 improvement over uniform alignment in Table 6 is rather marginal given the design complexity of the IRS-based weighting mechanism. It would be beneficial to supplement the paper with a task-level analysis examining whether IVA yields larger gains on specific task types where visual information dependency is high, such as TextVQA.


[W2]  TPA's empirical success under a theoretically limited one-step transition alignment lacks sufficient analysis.

TPA is a one-step transition alignment that theoretically cannot cover multi-step exposure bias, yet it shows empirical effectiveness — and this success remains unexplained. In all training scenarios of TPA, the prefix preceding the sampled token is always ground-truth, meaning the student never directly experiences prefixes where multiple positions simultaneously deviate from the reference sequence, which is precisely what occurs during actual inference. Nevertheless, Figure 6 demonstrates a substantial reduction in %ExAccErr. The paper does not investigate whether this improvement stems from one-step alignment generalizing to unseen prefixes, from benchmark responses being short enough that multi-step deviations never become severe, or from the model's inherently peaked output distribution limiting the degree of deviation in practice. Without such analysis, it is impossible to determine under what conditions TPA succeeds or fails, limiting confidence in the method's general applicability beyond the reported benchmarks.

---

> ### Author Rebuttal · Authors · 2026-03-30
>
> We sincerely appreciate your constructive comments and valuable suggestions. Thank you for the time and effort you have devoted to assessing our manuscript and helping us improve its quality. Our detailed responses to your concerns are provided below.
>
> ---
>
> > **W1/Q1: Is it possible to provide a task-level analysis to examine whether IVA yields larger gains on specific task types where visual information dependency is high, such as TextVQA?**
>
> We thank the reviewer for this insightful suggestion. As the reviewer suggested, we provide a task-level analysis from Table 5. **IVA yields the largest gains on tasks with high visual dependency: +2.2 on TextVQA and +2.0 on GQA.** TextVQA requires the model to read and understand text within images, and GQA demands compositional visual reasoning over scene graphs, both heavily relying on precise visual information extraction. This task-level pattern confirms that IVA works as designed: it transfers the teacher's instruction-relevant visual focus, directly benefiting tasks where accurate visual grounding is critical. We will add this analysis in the Component Analysis (Line 311-329) of the revised paper.
>
> |          | TextVQA     | GQA         |
> | -------- | ----------- | ----------- |
> | Baseline | 59.1        | 57.6        |
> | +IVA     | 61.3 (+2.2) | 59.6 (+2.0) |
>
> Regarding IVA's smaller average improvement compared to TPA. We acknowledge that IVA's average gain is more modest because it operates as an **indirect** supervision mechanism: rather than directly constraining the output distribution like TPA, IVA transfers the teacher's instruction-relevant visual focus by aligning visual token importance weights, implicitly guiding the student toward better visual grounding. This indirect nature inherently limits its average improvement magnitude. **However, we highlight that IVA is extremely lightweight and easy to integrate**: it introduces no additional learnable parameters, requires no extra forward passes, is easy to implement, and adds negligible overhead to total training time as shown below.
>
> | Method  | Training Time (H) |
> | ------- | :---------------: |
> | TPA     |        503        |
> | TPA+IVA |        509        |
>
> ---
>
> > **W2/Q2: Can you discuss the conditions under which the method is expected to generalize beyond the benchmarks?**
>
> We thank the reviewer for this insightful question. We agree that a deeper analysis of why one-step transition alignment effectively mitigates multi-step exposure bias is important. Below we provide both an in-depth explanation and a discussion on generalization.
>
> **Why one-step alignment can address multi-step exposure bias.** While TPA only explicitly aligns one-step transitions, its effect on multi-step exposure bias can be understood from two complementary perspectives. (1) **Expanded alignment scope**: as discussed in Section 3.2 Remark and Appendix D.1, Vanilla KD aligns along $O(|V|)$ generation paths, whereas TPA expands this to $O(|V|^2)$ by additionally aligning the transition matrix. This expanded coverage significantly increases the overlap between training-time prefixes (ground-truth) and inference-time prefixes (student-generated), reducing the accumulated error at each step. (2) **Error propagation suppression**: exposure bias arises from the accumulation of small per-step errors. By aligning the student's transition dynamics with the teacher's on the student's own on-policy samples, TPA ensures that even when the student deviates slightly, the subsequent predictions remain aligned with the teacher. This is empirically supported by Fig. 6, where TPA not only reduces the final %ExAccErr but also eliminates the growing trend observed in Vanilla KD, indicating that distributional divergence does not amplify with longer prefixes.
>
> **Generalization beyond the reported benchmarks.** TPA does not rely on any task-specific or benchmark-specific assumptions. Its core mechanism, on-policy transition probability alignment, is applicable to any autoregressive generation scenario. Empirically, Fig. 6 shows that %ExAccErr for TPA remains consistently low across all generation steps, without the growing trend observed in Vanilla KD, suggesting that TPA's effectiveness does not diminish as generation length increases. Since exposure bias is a fundamental challenge that worsens with longer sequences, we expect TPA to remain effective in broader and more complex generation settings. We also note that Reviewer 44W7 recognized that "the scaling analysis further demonstrates the method's generalization capability across heterogeneous model architectures."

---

> > ### Author Rebuttal · Reviewer_SvA6 · 2026-04-03
> >
> > Thank you for the detailed and thoughtful responses.
> >
> > For W1, I appreciate the additional task-level analysis. The results support the claim that IVA is particularly effective on tasks with high visual dependency, and the explanation is convincing.
> >
> > For W2, while the explanation is not purely theoretical, the empirical evidence and intuitive reasoning about generalization to be sufficiently comprehensive and helpful.
> >
> > Based on these clarifications, I will update my score accordingly.

---

### Official Review · Reviewer_1eec · 2026-03-12

**Soundness:** 3
**Presentation:** 2
**Significance:** 3
**Originality:** 3
**Overall Recommendation:** 4
**Confidence:** 3

**Summary:**

This paper presents a knowledge distillation (KD) framework for Multimodal Large Language Models (MLLMs) that moves beyond traditional next-token alignment by explicitly modeling token interactions. The key insight is that MLLMs rely on two critical interaction types: (1) vision-instruction token interactions for extracting relevant visual information during prefilling, and (2) intra-response token interactions for coherent autoregressive generation during decoding. The framework introduces two components: Instruction-aware Vision Alignment (IVA), which aligns visual tokens based on instruction-relevant importance weights to focus on salient regions, and Transition Probability Alignment (TPA), which aligns token-to-token transition probabilities to capture the teacher's dynamic generative logic. The authors propose an Instruction-Relevance Score (IRS) to identify optimal layers for visual alignment and a parallelized ribbon attention mechanism for efficient TPA computation. Experiments demonstrate that Align-TI achieves state-of-the-art results among compact MLLMs (1B-2B parameters).

**Compliance With Llm Reviewing Policy:**

Affirmed.

**Final Justification:**

Please see the rebuttal acknowledgement.

**Key Questions For Authors:**

Please refer to the strengths and weaknesses section.

**Strengths And Weaknesses:**

Strengths:
1.The paper is technically rigorous with well-motivated methodology.
2.The problem addressed -- efficient deployment of MLLMs on resource-constrained devices is highly relevant given the proliferation of edge AI applications.

Weaknesses:
1.While the paper mentions functional misalignment between student and teacher layers (line 71), it doesn't provide empirical analysis of how IVA/TPA interact with potential representation space mismatches at different depths.
2.While the paper distinguishes from concurrent MLLM KD methods, the relationship to broader sequence-level KD techniques could be more thoroughly discussed.
3.The paper would benefit from reporting any failed approaches attempted during development, e.g., did multi-layer IVA help? Was TPA tried with teacher-sampled tokens instead of student-sampled?

---

> ### Author Rebuttal · Authors · 2026-03-30
>
> We sincerely appreciate your constructive comments and valuable suggestions. Thank you for the time and effort you have devoted to assessing our manuscript and helping us improve its quality. Our detailed responses to your concerns are provided below.
>
> ---
>
> > **W1: It doesn't provide empirical analysis of how IVA/TPA interact with potential representation space mismatches at different depths.**
>
> We thank the reviewer for their careful review. Actually, **Align-TI is a token-level method that does not perform any layer-wise alignment**, and therefore does not suffer from representation space mismatches at different depths.
>
> Specifically, the "functional misalignment between student and teacher layers" mentioned in Line 71 was cited from prior work to explain why layer-wise feature alignment is ineffective for MLLM KD. Recognizing this exact limitation, recent MLLM distillation methods have shifted toward output-level alignment as a more effective paradigm. Our Align-TI follows this direction and does not perform any layer-wise alignment. Both IVA and TPA align solely on output distributions. While IVA utilizes attention maps, these serve strictly as weighting signals rather than intermediate alignment targets. Therefore, the representation space mismatch problem does not apply to our framework.
>
> ---
>
> > **W2: The relationship to broader sequence-level KD techniques could be more thoroughly discussed.**
>
> We appreciate the reviewer's suggestion.
>
> **First**, Align-TI is a MLLM KD framework modeled in view of token interactions: IVA distills the teacher's instruction-aware visual information extraction capability from vision-instruction token interactions, and TPA distills the teacher's dynamic generation capability from intra-response token interactions. Traditional sequence-level KD methods such as SeqKD[1] are designed for text-only language models and do not model these multimodal token interactions.
>
> **Second**, even comparing TPA with SeqKD directly, they differ fundamentally in both methodology and granularity. SeqKD is a **data-level** strategy that replaces the training data with teacher-generated sequences, requiring the teacher to autoregressively decode complete sequences for the entire training set. In contrast, TPA is an **objective-level** design that introduces an alignment loss on token-to-token transition probabilities, computed efficiently via a parallelized scheme without additional sequence generation. Moreover, we have provided both theoretical and empirical analysis showing that TPA effectively promotes sequence-level alignment in Appendix D.3.
>
> We will add the discussion of sequence-level KD in the "Knowledge Distillation for LLM" section of the Appendix A (Line 692-699). We thank the reviewer for helping us improve the paper.
>
> ---
>
>
> >**W3: The paper would benefit from reporting any failed approaches attempted during development.**
>
> We thank the reviewer for this suggestion. We share key negative results encountered during development:
>
> **Multi-layer IVA.** Table 7 shows that different layers provide drastically different supervision quality (e.g., Layer 7: 58.8 AVG vs. IRS-optimal Layer 21: 65.1). This suggests that aggregating across layers would dilute the precise focus from the optimal layer with noisy, less instruction-specific patterns. We will include these discussions in Line 350-369 of the revised paper.
>
> **Teacher-sampled TPA.** As discussed in Section 3.2, we considered conditioning on the teacher distribution instead of the student distribution. This was less effective for two reasons. First, it requires additional teacher forward passes for each sampled token, resulting in prohibitively expensive computational overhead. Second, teacher-sampled tokens are off-policy for the student: the teacher's high-probability tokens often overlap with the ground truth, providing redundant supervision, while failing to cover the student's own likely error modes where corrective alignment would be most valuable.
>
> ---
>
> [1] Kim, et al. "Sequence-level Knowledge Distillation." EMNLP. 2016.

---

> > ### Author Rebuttal · Reviewer_1eec · 2026-04-05
> >
> > NA

---

### Official Review · Reviewer_Vc7E · 2026-03-14

**Soundness:** 2
**Presentation:** 3
**Significance:** 2
**Originality:** 3
**Overall Recommendation:** 4
**Confidence:** 3

**Summary:**

The paper proposes Align-TI, a knowledge distillation (KD) framework for multimodal large language models (MLLMs). Motivated by the perspective of token interactions, the method consists of two components. First, Instruction-aware Vision Alignment (IVA) transfers the teacher’s instruction-relevant visual focus by aligning attention weights derived from the most instruction-relevant layer, selected using the proposed Instruction-Relevance Score (IRS). Second, Transition Probability Alignment (TPA) aligns one-step token transition probabilities, estimated with Monte Carlo sampling and an efficient parallelized computation scheme. Empirical results show that Align-TI consistently outperforms Vanilla KD and several prior baselines.

**Compliance With Llm Reviewing Policy:**

Affirmed.

**Final Justification:**

The rebuttal addressed most of my concerns. However, as other reviewers also pointed out, the performance gains are modest, and the methodological analysis is insufficient. I will maintain my score.

**Key Questions For Authors:**

* The paper would be better positioned with a more concrete methodological comparison to prior MLLM distillation methods such as Align-KD and LLaVA-MoD.
* For readability, it may be helpful to integrate Table 1 and Table 2.
* The paper provides qualitative evidence that IVA improves instruction-relevant visual focus. It would be great if such positive effect can be also quantitively supported.

**Limitations:**

yes

**Strengths And Weaknesses:**

**[Strengths]**
* The paper identifies an important limitation of prior MLLM distillation methods: static next-token alignment and off-policy KD. This contribution is clear and well-motivated.
* Both IVA and TPA are conceptually intuitive, and the ablation results support that each component contributes positively.
* The implementation of TPA appears practical with Monte Carlo approximation and a parallelized computation technique to reduce the cost of estimating transition probabilities. This makes the proposal more convincing from a systems perspective.
* Table 4 shows that the full Align-TI objective increases training time from 355 to 509 hours and memory from 70.6 to 75.6 GiB relative to Vanilla KD. Reporting this overhead explicitly is important, and I think this transparency is a strength of the paper.

**[Weaknesses]**
* The empirical advantage of the specific IVA design appears relatively modest. In Table 6, the gain over uniform visual-token alignment is only 64.5 vs. 65.1, and in Table 8, using teacher-derived importance weights improves only 64.7 vs. 65.1 over student-derived ones. These results suggest that, while instruction-aware weighting is beneficial, the practical margin of design details be limited.

---

> ### Author Rebuttal · Authors · 2026-03-30
>
> We sincerely appreciate your constructive comments and valuable suggestions. Thank you for the time and effort you have devoted to assessing our manuscript and helping us improve its quality. Our detailed responses to your concerns are provided below.
>
> ---
>
> > **W1: The empirical advantage of the specific IVA design appears relatively modest.**
>
> We thank the reviewer for this comment. IVA is designed to be simple but effective. We justify our specific design choices from two aspects:
>
> **(1) Consistent improvement.** Whether switching from uniform alignment to weighted alignment, or from student-derived weights to teacher-derived weights, our designs consistently improve performance across all six benchmarks. Beyond this, some benchmarks show notable gains, e.g., +1.7 on MME with instruction-aware weighting.
>
> **(2) Negligible additional cost.** Using weighted alignment instead of uniform alignment, and using teacher weights instead of student weights, do not require any additional forward passes. The teacher already computes and caches its attention states during its forward pass, so extracting these weights is virtually free and requires only a few lines of code to implement. As a whole, IVA adds only 1.1% overhead to total training time (Table 4, Line 303-307).
>
> Thus, obtaining a more effective distillation signal and consistent performance gains at negligible extra cost firmly justifies our design choices.
>
> ---
>
> > **Q1: It would be beneficial to include a more concrete methodological comparison to prior MLLM distillation methods such as Align-KD and LLaVA-MoD.**
>
> We appreciate the reviewer's suggestion.
>
> **Align-KD[1]** focuses on **cross-modal feature-level representation alignment**. It addresses the alignment problem by distilling how the teacher projects vision tokens into the text embedding space, utilizing attention distillation to guide the student's cross-modal matching at shallow layers.
>
> **LLaVA-MoD[2]** focuses on **architectural enhancement**. It tackles the student's representational capacity bottleneck by **incorporating MoE layers**, which is a structural modification rather than an improvement to the distillation objective itself.
>
> **In contrast, our Align-TI** focuses on **output-level distillation**, extracting the knowledge embedded in the dynamic output token interaction. Beyond simply imitating the teacher's final output tokens, Align-TI teaches the student the teacher's reasoning process: how instruction and visual tokens interact to extract relevant information (IVA), and how tokens transition during autoregressive generation (TPA).
>
> We thank the reviewer again for this valuable suggestion, and we will incorporate this comprehensive comparison into the revised related work section (Line 700-710).
>
> ---
>
>
> > **Q2: For readability, it would be helpful to integrate Table 1 and Table 2.**
>
> Thank you for this suggestion. We agree that integrating the teacher model performance into the main comparison table would provide readers with a more direct and complete view. We will merge Table 1 into Table 2 in the revised version.
>
> ---
>
>
> > **Q3: It would be great if the positive effect of IVA on visual focus could be quantitatively supported.**
>
> We thank the reviewer for this constructive suggestion. To quantitatively validate that IVA improves instruction-relevant visual focus, we compute the cosine similarity between the student's and teacher's visual token importance weights at the IRS-optimal layer. Since these importance weights directly quantify how much attention the model allocates to each visual token, a higher similarity indicates that the student has more accurately acquired the teacher's instruction-driven visual focus. We evaluate this metric on the validation sets of the six benchmarks for models trained with and without IVA. The results are as follows:
>
> | Model   | GQA   | SQA   | TextVQA | POPE  | MME   | MMB   | AVG   |
> | ------- | ----- | ----- | ------- | ----- | ----- | ----- | ----- |
> | w/o IVA | 0.660 | 0.261 | 0.719   | 0.655 | 0.654 | 0.702 | 0.609 |
> | w/ IVA  | 0.677 | 0.300 | 0.753   | 0.666 | 0.691 | 0.764 | 0.642 |
>
> As shown in the table, equipping the student with IVA significantly increases the similarity of its visual token importance weights to those of the teacher across all benchmarks. Combined with the qualitative visualization in Fig. 5, these results provide strong quantitative evidence that IVA successfully transfers the teacher's instruction-relevant visual focus to the student. We will add this experiment to the revised manuscript (Line 400-415). We deeply appreciate the reviewer's guidance in improving the quality of our paper.
>
> ---
>
> [1] Feng, Qianhan, et al. "Align-KD: Distilling Cross-Modal Alignment Knowledge for Mobile Vision-Language Large Model Enhancement." CVPR, 2025.
>
> [2] Shu, Fangxun, et al. "LLaVA-MoD: Making LLaVA Tiny via MoE Knowledge Distillation." ICLR, 2025

---

> > ### Author Rebuttal · Reviewer_Vc7E · 2026-04-02
> >
> > The authors addressed most of my questions. I'd maintain my score.

---

> > > ### Author Response · Authors · 2026-04-07
> > >
> > > We sincerely thank Reviewer Vc7E for the valuable feedback throughout the review process, which has helped us improve the quality of our paper. We notice that the reviewer raised a new point in the final justification regarding performance gains and methodological analysis, and we would like to provide a brief clarification.
> > >
> > > **Regarding performance gains.** Align-TI achieves +2.4 average improvement over SFT and consistent gains over Vanilla KD across all six benchmarks. For TPA, it contributes +2.1 AVG improvement and effectively suppresses exposure bias (Fig. 6). For IVA, it yields notable improvements on vision-dependent tasks (e.g., +2.2 on TextVQA and +2.0 GQA) at only 1.1% additional training cost. Reviewer SvA6, who originally raised this concern, acknowledged that the task-level analysis is convincing.
> > >
> > > **Regarding methodological analysis.** Regarding methodological analysis, we have provided substantial analysis in the current manuscript and rebuttal, including the in-depth discussion of alignment scope expansion (Section 3.2 Remark, Appendix D.1), the connection between TPA and sequence-level alignment (Appendix D.3), and empirical validation of error propagation suppression. Reviewer SvA6, who originally raised this concern, acknowledged that the analysis is sufficiently comprehensive and helpful and updated their score accordingly.

---

### Decision · Program_Chairs · 2026-04-30

**Decision:**

Accept (regular)

**Comment:**

This paper introduces Align-TI, a knowledge distillation framework for multimodal large language models that explores token interactions beyond traditional next-token alignment. The method consists of two core components. First, Instruction-aware Vision Alignment transfers the teacher model's visual focus by assigning importance weights to visual tokens based on instruction-to-vision attention maps. This process uses an Instruction-Relevance Score to identify the optimal layer for alignment. Second, Transition Probability Alignment aligns one-step token transition probabilities to capture the teacher's dynamic autoregressive generation patterns and mitigate exposure bias. The framework utilizes Monte Carlo sampling and a parallelized ribbon attention mechanism for efficient computation. Experiments show that Align-TI achieves state-of-the-art results among 1B to 2B parameter models, with the 2B model surpassing LLaVA-1.5-7B by a relative 7.0%.

This paper receives all positive scores and is recommended for acceptance. The reviewers found the proposed method well-motivated and supported by extensive experiments and ablation studies. During the rebuttal phase, the authors successfully addressed the reviewers' concerns. Reviewer SvA6 revised their score upward because the additional ablation study and explanations clarified their questions regarding one-step transition alignment. Reviewer 1eec noted their concerns were fully resolved. Reviewers Vc7E and 44W7 maintained their positive scores after the rebuttal. Although Reviewer Vc7E mentioned that the performance gains were modest and the methodological analysis was somewhat insufficient, all reviewers reached a consensus. Therefore, I recommend accepting this paper.